# The Roles of Dissociation and Depression in PTSD Among Soldiers Exposed to Combat

**DOI:** 10.3390/ijerph22060814

**Published:** 2025-05-22

**Authors:** Leah Shelef, Nir Spira, Uzi Bechor, Jacob Rotschield, Eran Shadach

**Affiliations:** 1School of Social Work, Sapir College, D.N. Hof Ashkelon 7916500, Israel; 2Center for Mental Health Services and Mental Resilience, Israel Defense Forces-Medical Corps, Ramat Gan 5262000, Israeljacobrotschield@gmail.com (J.R.); 3School of Social Sciences, The Academic College of Tel Aviv-Yaffo, Tel Aviv Yaffo 6818211, Israel; nir.spira@gmail.com (N.S.); eransh@mta.ac.il (E.S.)

**Keywords:** PTSD, depression, dissociation, military, post-traumatic stress symptoms

## Abstract

Exposure to severe combat situations significantly raises the risk of depression and post-traumatic stress disorder (PTSD). Trauma survivors may use dissociation as a defense mechanism, increasing the likelihood of PTSD. This study aims to explore the roles of dissociation and depression in PTSD among soldiers exposed to combat who sought help from the Israel Combat Stress Reaction Unit. **Method:** This cross-sectional study involved 927 individuals who participated in a particular military operation in 2014 [98.5% male (*n* = 906); mean age = 27.08 (SD = 5.93)]. Participants completed three questionnaires: the Dissociative Experiences Scale (DES), the Beck Depression Inventory (BDI), and the Post-Traumatic Stress Symptom Checklist (PCL-5) for PTSD. **Results:** Our results showed that severe PTSD (PCL score ≥ 33) was found in 30.4% of participants, and 76.6% showed dissociative symptoms (DES score ≥ 30). Additionally, 23.5% experienced moderate depression, while 19.1% reported severe depressive symptoms. A Generalized Linear Model revealed that both depression and dissociation significantly contribute to PTSD. Individuals with depression were three times more likely to experience post-traumatic symptoms compared to 1.23 times for those with dissociative symptoms. **Conclusions:** Life-threatening situations significantly predicted higher PTSD symptoms, serving as a risk factor for depression and dissociation, which play important roles in PTSD, with depression having notably greater impact.

## 1. Introduction

Exposure to combat increases risk for post-traumatic stress disorder (PTSD) [1]. The DSM-5 PTSD definition comprises a variety of symptoms that fall into four clusters: re-experiencing, avoidance, negative alterations in cognition and mood, and alterations in arousal and reactivity [2]. Exposure to combat may involve a range of experiences whose association with mental health outcomes varies. For example, taking a life is an event that strongly predicts PTSD and depressive symptoms [1]. Research has revealed a connection between PTSD and individuals who have witnessed dead bodies or human remains, particularly highlighting its relation to the Intrusion cluster (i.e., re-experiencing) of symptoms. These findings enhance our understanding of the various clusters associated with PTSD, especially those that have a strong correlation with the severity of PTSD symptoms [3,4].

PTSD frequently co-occurs with major depressive disorder [5], with complex bidirectional relationships involving shared symptoms (e.g., low mood, loss of interest or pleasure, feelings of guilt, sleep disturbances, low energy, and recurrent thoughts of death), overlapping risk factors, and potential mutual causal pathways [5,6,7,8]. Regarding PTSD, research shows various patterns—depression preceding PTSD [9], PTSD predicting depression [10], PTSD mediating between combat exposure and depression [11], or both disorders emerging from common risk factors [12,13]. Moreover, soldiers who were exposed to combat subsequently exhibited significantly more depressive symptoms compared to those who were deployed but not exposed to combat, as well as those who were not deployed at all [14]. Understanding these interrelationships is crucial for effective treatment [15]. Moreover, understanding the relationship between PTSD and depression is crucial, as their comorbidity represents a significant risk factor for suicidal behavior among military personnel [16], as well as completed suicides within military populations [17].

PTSD often includes dissociative symptoms [18,19], recognized as a specifier in DSM-5-TR [20] and a multidimensional psychopathology marker [21]. While some studies show no PTSD–dissociation link [22], others correlate higher dissociation with greater PTSD severity [19,23]. Dissociation manifests as disruptions in psychological process integration, appearing as depersonalization, bodily/mental detachment, or environmental unreality perceptions [2]. Functioning as a defense mechanism, it impairs integration of experiences with emotions, providing protection from overwhelming situations [24]. While initially maintaining self-continuity during trauma, dissociation can lead to fragmentation of aspects of self [25] and eventually become a generalized response triggered by minor stressors [26,27]. Among Israeli veterans, dissociation and somatization are key risk factors for developing full PTSD [28].

In relation to suicide, dissociative symptoms predicted suicide risk above other comorbidities and trauma history among U.S. veterans. Moreover, dissociative symptoms in veterans may be a transdiagnostic risk factor independent of PTSD [29]. Additionally, dissociation has been linked to suicide attempts among Israeli soldiers—not directly correlated with stress or psychopathologies, but potentially functioning as a facilitating mechanism for suicidal behavior under extreme stress [30]. This relationship warrants further investigation, particularly in combat-related PTSD.

Research has also identified a connection between dissociation and depression [31]. In a study that compared participants (*N* = 410) from 18 different countries with low dissociative symptoms to those with high dissociative symptoms, the latter group reported more depressive symptoms [32]. Furthermore, it has been suggested that depression might obscure pathological dissociation, which could lead to misdiagnosis and a lack of targeted treatments for dissociative processes [33]. These findings emphasize the necessity of personalized treatments that target both depressive symptoms and dissociation.

In summary, dissociative responses to trauma can increase the risk of developing PTSD [19] and depression [31]. Research has shown a complex relationship where dissociation can act as a mediating variable, exacerbating the symptoms of both PTSD and depression [34]. Studies have demonstrated that higher levels of dissociation are linked to more severe symptoms of PTSD and depression in civilian populations [32,35] as well as among police officers [36]. Similarly, investigations involving active U.S. service members and veterans with PTSD [34] have found a positive association between dissociation and the development of depression over time. These findings highlight the importance of early identification and targeted treatment of dissociative symptoms in individuals with PTSD and depression, particularly among veterans, to improve therapeutic outcomes and prevent further psychological decline.

Researching complex interrelations of these variables is crucial during the October 2023 war (ongoing as of the time of writing), as mental health support needs have surged [37]. Post-October 7th studies show high prevalence rates: PTSD (29.8–31.4%), depression (42.7%), and anxiety (44.8%) [38,39,40], with predictions that 5% of the Israeli population will develop PTSD [41]. Given limited data on current soldiers’ mental health, examining those who served in a military operation in 2014 and sought help from the Israel Defense Forces (IDF) Combat Stress Reaction Unit (CSRU) offers valuable insights for improving early identification and psychological care. The CSRU was established within the Israeli military to treat military veterans with PTSD and other stress-related disorders resulting from their military service. All data used in this study, including self-report questionnaires, were collected during the intake meeting upon admission to the unit. The CSRU employs various therapeutic approaches, including trauma-focused group interventions, prolonged exposure therapy, eye movement desensitization and reprocessing (EMDR), and cognitive-behavioral therapy (CBT). Additionally, the CSRU offers instruction and counseling for IDF mental health professionals.

### The Present Study

This study aims to examine the unique contributions of depression and dissociation to PTSD among Israeli veterans who participated in a military operation in 2014. Investigating this issue is particularly important in light of the ongoing war and its psychological impact, which has resulted in a significant increase in the number of individuals seeking mental health support. The findings of this study may enhance our understanding of the complex interplay between PTSD, depression, and dissociation and contribute to the early identification of complex cases, ultimately improving the quality of therapeutic responses for those seeking help. Our hypotheses were as follows:
A significant positive relationship will be found between depression and post-traumatic symptoms; soldiers experiencing higher levels of depression will also exhibit greater post-traumatic symptoms.A positive relationship will be found between dissociation and PTSD, such that soldiers who experience higher levels of dissociation will also show higher levels of post-traumatic symptoms.Both depression and dissociation contribute to PTSD, with depression having a greater impact than dissociation.

## 2. Materials and Methods

### 2.1. Study Participants and Procedure

This study sample is part of a larger dataset derived from the medical records of 1642 male outpatient veterans who sought help at the CSRU between 1 January 2014 and 23 May 2023 [secondary data] [42]. The inclusion criteria required participants to have taken part in the military operation in 2014; therefore, only 927 veterans were included in the study. It is important to note that all participants experienced war-related trauma during their military service, which is a criterion for admittance to the outpatient unit. As a result, their stress-related disorders were linked to their military service. See Table 1 for descriptive statistics.

### 2.2. Measures

The background information includes the following details: the individual’s age at the time of admission for evaluation, gender (male/female/other), country of birth (Israel/other), and marital status (single/married/other). Military service-related variables are as follows: service type (combat support/combat), whether the individual was an officer in the military (yes/no), and an intellectual assessment score that evaluates general cognitive abilities (hereinafter: IQ). This score is measured on a 9-point scale ranging from 10 (very low) to 90 (very high) in 10-point increments. Additionally, information regarding whether the individual experienced a life-threatening situation (yes/no) is included.

Dissociative Experience Scale—DES [43,44]: This is a self-report questionnaire designed to examine the intensity and frequency of the dissociative symptoms in an individual. The questionnaire includes 28 items on a Likert-type scale ranging from 0% (Never) to 100% (Always). The average score for the total of all 28 items determines the dissociation rate. The cutoff point is an average score of 30% of the items, pointing to a high level of dissociation [43]. Cronbach’s alpha in the current study for PTSD was α = 0.95.

Depressive symptoms were measured using the Beck Depression Inventory—BDI [45]. The questionnaire comprises 21 items measured on a 4-point Likert scale ranging from 0 to 3, with a total score of 0–63. The internal reliability of the total BDI score was good (Cronbach’s α = 0.90). Cronbach’s alpha in the current study for depressive symptoms was α = 0.91.

PTSD symptoms were measured using the PTSD Checklist—PCL-5 [46]. This questionnaire includes 20 items relating to the four DSM-5 PTSD clusters: re-experiencing, avoidance, negative alterations in cognition and mood, and alterations in arousal and reactivity. Items are scored on a 5-point Likert scale ranging from 0 (not at all) to 4 (extremely). The total symptom severity score ranges from 0 to 80. A provisional PTSD diagnosis is suggested when the total score is equal to or exceeds a score of 33 [47]. The internal reliability of the PCL total score was good (Cronbach’s α = 0.93).

### 2.3. Statistical Analysis

Data were analyzed using SPSS version 29 (Armonk, NY, USA: IBM Corp). We used descriptive statistics to present the demographics and military service-related variables and the main study variables of PTSD, dissociation, and depression [means and standard deviations for continuous variables and distributions for the categorical variables (%, N)]. Then, we used Spearman’s correlations to examine the association between the three variables (PTSD, dissociation, and depression) and hypotheses 1–2.

Then, to analyze PTSD dissociation and depression in different background variables and military service-related ones, we used the Mann–Whitney (Wilcoxon W) test for continuous parametric distribution (i.e., age, intellectual assessment score). Finally, to examine hypothesis 3, which examines the relationship between dissociation, depression, and PTSD, we conducted a Generalized Linear Model to determine which variable had a more significant impact on PTSD.

### 2.4. Ethics

The Institutional Review Board of the IDF Medical Corps approved the study and waived the requirement for informed consent (No. 2269-2021).

## 3. Results

Descriptive statistics of the study variables are presented in Table 1. As can be seen in Table 1, among the participants, 69.6% (n = 610) were found to have probable PTSD, as indicated by PCL-5 scores of 33 or higher. Additionally, 23.4% of participants reported dissociative symptoms that also exceeded the clinical threshold (DES ≥ 30%). Regarding depression, 23.5% exhibited symptoms of moderate depression, while 19.1% reported symptoms of severe depression. Additionally, the mean score for all participants indicated severe post-traumatic stress, with a mean of 42.51 (SD = 18.80). In contrast, the average level of dissociation for the entire sample was below the threshold index, with a mean of 20.06% (SD = 17.57). Regarding depression, the overall average of all participants showed mild depression, with a mean of 18.64 (SD = 10.84).

Spearman’s correlation revealed a significant positive association between the three variables: PTSD (i.e., PCL), dissociation (i.e., DES), and depression (i.e., BDI). The higher the level of dissociation, the higher the level of PTSD (r = 0.660; *p* < 0.001). Similarly, the higher the level of depression, the higher the level of PTSD (r = 0.788; *p* < 0.001).

A significant negative relationship was identified between intelligence and three key study variables: post-traumatic symptoms, dissociation, and depression. The correlations were as follows: intelligence and PTSD (r = −0.119; *p* = 0.001), intelligence and dissociation (r = −0.113; *p* = 0.003), and intelligence and depression (r = −0.125; *p* = 0.003). In contrast, positive relationships were observed between age and PTSD (r = 0.105; *p* < 0.02), age and dissociation (r = 0.073; *p* = 0.039), and age and depression (r = 0.125; *p* = 0.002).

The relationship between the background variables and military service-related and between PTSD, dissociation, and depression is presented in Table 2. In examining the association between demographic and service-related variables and post-traumatic stress symptoms, only military rank (i.e., not officer) and exposure to a life-threatening situation showed significance. Non-officers reported higher average PTSD symptoms (*p* = 0.012) and those who experienced trauma had more symptoms than those who did not (*p* = 0.003). Likewise, non-officers also had a higher average for depressive symptoms (*p* = 0.023).

The first two hypotheses proposed that there would be a positive relationship between depression and post-traumatic symptoms, as well as between dissociation and post-traumatic symptoms. Spearman’s correlation analysis indicated a statistically significant relationship between depression and post-traumatic symptoms (r = 0.788; *p* < 0.001) and a significant relationship between dissociation and post-traumatic symptoms (r = 0.660; *p* < 0.001). These results confirm both hypotheses.

To analyze the third research hypothesis, which posits that both depression and dissociation contribute to PTSD, with depression having a more significant impact than dissociation, we conducted a Generalized Linear Model regression, as shown in Table 3. This analysis aimed to assess the relative contributions of dissociation and depression to PTSD. In the first regression model, which did not include the depression and dissociation variables, we found it significant (R2 = 25.707; *p* = 0.001). The life-threatening variable also showed significance (*p* = 0.007), indicating that the absence of life-threatening served as a protective factor (1:0.18) times 55.5. Furthermore, the results indicated that as age increases, the severity of post-traumatic stress also increases by 1.4 times for each additional year (*p* = 0.010). Intellectual assessment score (i.e., IQ) was identified as another protective factor; as intelligence levels rise, the risk associated with PTSD decreases (*p* = 0.014).

We found the model significant in a second regression analysis (R2= 495.559; *p* < 0.001). Both depression and dissociation were significant predictors of post-traumatic stress. Specifically, for every unit increase in the Post-Traumatic Checklist (PCL), depression increased threefold, while dissociation increased by 1.23-fold.

Regarding our third research hypothesis—that both depression and dissociation would contribute to PTSD, with depression having a higher contribution than dissociation—we confirmed this hypothesis. In conclusion, our findings indicate that depression has a more significant impact on post-traumatic stress than dissociation.

## 4. Discussion

The present study examined the relationship between PTSD, depression, and dissociation among veterans who sought help at the CSRU following their participation in the military operation in 2014. Our findings support all three hypotheses, demonstrating significant positive correlations between depression and PTSD symptoms, between dissociation and PTSD symptoms, and confirming that both depression and dissociation contribute to PTSD, with depression having a greater impact.

Our results revealed that approximately 70% of participants experienced probable PTSD, as indicated by PCL-5 scores above the clinical threshold. This prevalence rate is substantially higher than those reported in previous studies among military populations, which typically range from 8 to 23% [6,48]. This disparity is likely explained by our sample consisting entirely of treatment-seeking veterans, as opposed to general military populations examined in other studies.

Regarding the relationship between depression and PTSD, our findings align with the existing literature, which has consistently documented a strong association between these disorders [5,10]. The robust correlation we observed (r = 0.788) supports the notion of a complex interrelationship between these conditions, potentially involving shared vulnerability factors, overlapping symptoms, or causal pathways [7,8]. This relationship is particularly significant given that comorbid PTSD and depression represents a heightened risk factor for suicidal behavior among military personnel [16,17].

Similarly, the significant relationship between dissociation and PTSD (r = 0.660) supports previous research indicating that higher levels of dissociation correlate with greater PTSD severity [19,23]. This finding is consistent with the recognition of dissociation as a specifier for PTSD in the DSM-5-TR [24] and reinforces the understanding of dissociation as a defense mechanism that may initially serve a protective function but can ultimately contribute to more severe psychopathology [26,27].

Our regression analysis further revealed that both depression and dissociation significantly predict PTSD symptoms, with depression exerting a stronger influence. Specifically, individuals with depression were three times more likely to experience post-traumatic symptoms compared to 1.23 times for those with dissociative symptoms.

Interestingly, our study also identified several demographic and military service-related factors associated with psychological distress. The negative correlation between intelligence and all three study variables (PTSD, dissociation, and depression) aligns with research suggesting that higher cognitive abilities may serve as a protective factor against psychopathology following trauma exposure [49].

Our results confirmed that exposure to life-threatening situations significantly predicted higher PTSD symptoms, serving as a risk factor for psychological distress. This finding is consistent with dose–response models of trauma, which posit that more severe or threatening traumatic experiences are associated with greater psychological impact [1].

The innovation of this study is that it emphasizes that the role of dissociation in PTSD is complex and not straightforward. The existing research landscape on this is mixed, with, some studies finding no significant relations between the two, while others have found some. In this regard, the current results support a connection. Moreover, dissociation has also been found to correlate with suicidal behavior among soldiers, emphasizing the importance of the current results establishing the relationship between PTSD, dissociation, and suicidality. Furthermore, considering the dual role of dissociation as both a protective mechanism and a potential risk factor, the current findings shed light on the latter, suggesting that beyond a certain threshold, it can transition into a maladaptive response, increasing vulnerability to severe psychopathology.

## 5. Limitations and Implications

The findings indicate several important clinical implications for assessment and treatment of combat-exposed veterans that arise from the results of the study as well as its limitations. First, the cross-sectional design of the study limits our ability to draw causal conclusions about the relationships among PTSD, depression, and dissociation. Given the strong interconnections among these three conditions, it is essential to conduct comprehensive evaluations addressing them. Clinicians should be aware that the presence of one condition increases the likelihood of the others, making thorough screening and ongoing monitoring essential. Moreover, it is crucial to explore risk factors in greater depth, particularly the symptom overlap between PTSD, depression, and dissociation. Second, another implication in this context is the study’s finding that depression significantly impacts PTSD more than dissociation suggests that focusing on depressive symptoms may be particularly beneficial for treating combat-related PTSD. This emphasizes the need for personalized treatment plans that address both depressive symptoms and dissociation. Third, the sample in this study consisted only of treatment-seeking male veterans, which limits the generalizability of the results to female veterans, non-treatment-seeking populations, or civilians who have experienced trauma. Future research should investigate these relationships in more diverse populations. Finally, it is crucial to train mental health professionals to better understand the relationships among these three disorders and to implement tailored clinical assessments and therapeutic approaches.

Of note, this is the first study to examine the topic among a military population in Israel, giving the findings timely relevance in context of the ongoing war. While this novelty may include a limitation regarding universal generalizability, various characteristics of this demographic (e.g., mandatory service, current war) merit further research for similar military populations and frameworks.

## 6. Conclusions

In conclusion, this study advances our understanding of the complex interrelationships among PTSD, depression, and dissociation in combat-exposed veterans. The findings highlight the significant impact of depression on PTSD and identify several important risk and protective factors.

Our findings are aligned with the existing literature, pointing to potentially shared vulnerability factors, overlapping symptoms, or causal pathways, and reinforce the understanding of the positive and negative implications of dissociation as a defense mechanism. Specifically, life-threatening situations significantly predicted higher PTSD symptoms, serving as a risk factor for depression and dissociation, which play important roles in PTSD, with depression having notably greater impact. As discussed above, the findings offer several important and timely practical and theoretical implications for clinicians, training, and research.

## Figures and Tables

**Table 1 ijerph-22-00814-t001:** Descriptive statistics of the study variables (N = 927).

		*n*	*%*	M	SD	Median	Minimum	Maximum
**Gender**	Male	906	98.5					
	Female	14	1.5					
**Country of birth**	Israel	805	90.7					
	Other	83	9.3					
**Marital status**	Single	684	77.7					
	Married	197	22.4					
**Service type**	Combat support	76	8.7					
	Combat	796	91.3					
**Officer in the military service**	Yes	104	11.7					
	No	786	88.3					
**Life-threatening**	No	297	36.7					
	Yes	512	63.3					
**Age**		882	95.14	27.082	5.932	26.0	19.0	79.0
**IQ**		743	80.15	57.47	17.462	60.0	10.0	90.0
**PCL**	Up to 33	266	30.4					
	33 and above	610	69.6					
**DES**	Up to 30%	629	76.6					
	31% and above	192	23.4					
**BDI**	No	400	64.3					
	Yes	222	35.7					
**BDI**	Normal (0–13)	211	33.9					
	Mild (14–19)	146	23.5					
	Moderate (20–28)	146	23.5					
	Severe (29–63)	119	19.1					
**PCL mean**		876	94.8	42.510	18.797	44.0	0	80
**DES mean**		821	88.56	20.055	17.565	15.3	0	92.14
**BDI mean**		622	67.09	18.641	10.840	17.0	0	57

**Table 2 ijerph-22-00814-t002:** The relationship between background variables and PTSD, dissociation, and depression.

			*n*	M	SD	Median	*z*	*p*-Value
**PCL**	**Gender**	Male	857	42.445	18.744	44.0		
		Female	13	49.076	18.834	50.0	−1.173	0.241
	**Country of birth**	Israel	777	42.522	18.740	44.0		
		Other	81	43.469	19.343	45.0	−0.446	0.656
	**Marital status**	Single	655	42.717	19.011	45.0		
		Married	193	41.559	17.897	41.0	−1.152	0.249
	**Service type**	Combat support	76	45.236	17.119	49.0		
		Combat	765	41.992	18.799	44.0	−1.414	0.157
	**Rank**	Other ranks	759	43.164	18.584	45.0		
		Officer	100	38.230	18.909	37.0	−2.504	0.012
	**Life-threatening**	No	297	39.872	18.970	42.0		
		Yes	512	44.046	18.218	46.0	−3.017	0.003
**DES**	**Gender**	Male	806	20.084	17.546	15.3		
		Female	9	17.817	17.483	10.7	−0.439	0.661
	**Country of birth**	Israel	730	20.334	17.787	15.3		
		Other	74	18.463	15.573	14.3	−0.708	0.479
	**Marital status**	Single	617	20.062	17.737	15.3		
		Married	177	19.514	16.884	14.3	−0.075	0.940
	**Service type**	Combat support	65	22.896	18.457	18.6		
		Combat	721	19.486	17.245	14.6	−1.462	0.144
	**Rank**	Other ranks	711	20.360	17.495	15.7		
		Officer	92	17.690	17.702	11.8	−1.876	0.061
	**Life-threatening**	No	282	18.686	16.381	13.9		
		Yes	474	20.599	18.210	16.1	−1.191	0.234
**BDI**	**Gender**	Male	609	18.633	10.780	17.0		
		Female	7	23.000	17.097	13.0	−0.321	0.749
	**Country of birth**	Israel	558	18.713	10.781	17.5		
		Other	49	19.449	12.020	17.0	−0.136	0.892
	**Marital status**	Single	500	18.850	11.020	18.0		
		Married	104	18.076	10.185	16.0	−0.724	0.469
	**Service type**	Combat support	47	19.446	10.195	19.0		
		Combat	564	18.540	10.853	17.0	−0.570	0.569
	**Rank**	Other ranks	545	18.979	10.749	18.0		
		Officer	60	16.050	10.822	14.0	−2.277	0.023
	**Life-threatening**	No	219	17.707	9.928	17.0		
		Yes	385	19.226	11.174	17.0	−1.231	0.218

**Note**: Mann–Whitney test.

**Table 3 ijerph-22-00814-t003:** Analyses of the relative impact of dissociation and depression on predicting post-traumatic symptoms.

		B	Wald Chi-Square	*df*	Exp (B) (OR)	95% C.I. for EXP (B)		*p*-Value
						Lower	Upper	
Constant		32.323	13.617	1	1.091 × 10^14^	3,816,831.321	3.116 × 10^21^	<0.001
Gender	Male	−2.121	0.123	1	0.120	8.455 × 10^−7^	17,019.232	0.726
	Female	0^a^			1			
Country of birth	Israel	0.793	0.097	1	2.209	0.015	327.622	0.756
	Other	0^a^			1			
Marital status	Single	3.754	3.382	1	42.692	0.781	2333.515	0.066
	Married	0^a^			1			
Service type	Combat support	5.002	3.867	1	148.717	1.017	21,753.640	0.049
	Combat	0^a^			1			
Rank	Other ranks	5.060	4.785		157.607	1.693	14,675.839	0.029
	Officer	0^a^			1			
Life-threatening	No	−4.027	7.236	1	0.018	0.001	0.335	0.007
	Yes	0^a^			1			
Age		0.353	0.136	1	1.423	1.090	1.859	0.010
IQ		−0.088	0.043	1	0.916	0.842	0.997	0.042
**GLM 1**: Likelihood Ratio Chi-Square = 25.707; *p* = 0.001								
Constant		13.421	3.720	1	673,993.415	0.805	5.644 × 10^11^	<0.001
Gender	Male	−0.306	0.003	1	0.736	2.658 × 10^−5^	20,389.097	0.953
	Female	0^a^						
Country of birth	Israel	1.647	0.759	1	5.192	0.128	211.059	0.384
	Other	0^a^						
Marital status	Single	0.329	0.044	1	1.390	0.064	30.285	0.834
	Married	0^a^						
Service type	Combat support	2.703	1.857	1	14.919	0.306	727.742	0.173
	Combat	0^a^						
Rank	Other ranks	2.359	1.856	1	10.576	0.355	314.807	0.173
	Officer	0^a^						
Life-threatening	No	−1.888	3.156	1	0.151	0.019	1.215	076
	Yes	0^a^						
Age		−0.042	0.149	1	0.959	0.774	1.187	0.699
IQ		0.014	0.198	1	1.014	0.953	1.079	0.656
Dissociation		0.212	29.506	1	1.237	1.145	1.335	<0.001
Depression		1.136	312.908	1	3.115	2.747	3.533	<0.001
**GLM 2**: Likelihood Ratio Chi-Square = 495.559; *p* < 0.001								

***Note***: Generalized Linear Models; 0^a^ Set to zero because this parameter used as reference.

## Data Availability

Data supporting the findings of this study are available on request from the corresponding author.

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
