# Peer review of "The Roles of Dissociation and Depression in PTSD Among Soldiers Exposed to Combat"

_ijerph, 2025, doi:10.3390/ijerph22060814_

Round 1

Reviewer 1 Report

Comments and Suggestions for Authors

I was very interested in reading this paper and, therefore, I thank the authors for submitting it to this journal and the editor for indicating me as a potential reviewer. Objectively, the paper does not shine for originality, however it offers some interesting ideas and the rigorous methodology, the large sample and the geographical context lead me to think that the study deserves to be shared with the scientific community. Here are my suggestions for an important revision of the paper:

- first of all, it is necessary to introduce the concept of trauma and PTSD. Although they are well-known concepts, I think the beginning of the paper is rude.
- when discussing trauma and PTSD, especially in the military context, it could be useful to indicate what and which elements most frequently trigger PTSD symptoms in this population. Furthermore, the topic is not only connected to the risk of death and the violence suffered, but the military also have to deal with the topic of being the authors of the death of others, and this could be a source of PTSD to take into account (https://doi.org/10.1002/jts.22630).
- Explore more closely the topic of the relationship between suicide risk and dissociation. Explain better why and how the two constructs are associated with each other.
- better focus on the differential diagnosis: depression/dissociation.
- better describe the specific tasks and functions of the CSRU.
- However, the authors seem to ignore the real limitations of the research. Dedicate a specific paragraph to the limitations of the research and the implications for future research.
- The practical implications of the research are very limited and should instead be more extensive and in-depth, dedicating a specific section to them before the conclusion. I expect that the practical implications will deal with psychotherapy, clinical assessment and prevention, as well as the training of mental health workers.

Reviewer 2 Report

Comments and Suggestions for Authors

In general, the text of the article is logically structured. The data obtained in the study are sufficiently highlighted and discussed. Correct conclusions are drawn.

The main flaw of the article is the lack of clear scientific novelty. The convincing statement of the impact of hostilities on the mental state of Israeli servicemen is of unquestionable value, but it does not contain significantly new information about the connections between PTSD, depression and dissociation. The specific numerical indicators of the greater importance of depression than dissociation in causing PTSD attract attention, but can they be recognized as universal properties? Perhaps the authors should focus more clearly on demonstrating the novelty of their results.

In section 2.1. Study Participants and Procedure, it seems appropriate to move the second and third paragraphs (which contain a description of the functions of the CSRU and the features of the recruitment of new recruits to the Israel Defense Forces) to the first paragraph – after the mention of the CSRU.

Instead, it is better to present the quantitative characteristics of the sample according to the measures used in the study below – after the description of these measures.

It is also necessary to check the correspondence of the numerical data of the Generalized Linear Model regression in the text and in the Table 3.

Round 2

Reviewer 1 Report

Comments and Suggestions for Authors

Thank you!!!